# Microbiological Contamination of Strawberries from U-Pick Farms in Guangzhou, China

**DOI:** 10.3390/ijerph16244910

**Published:** 2019-12-05

**Authors:** Xiaohong Wei, Shuiping Hou, Xinhong Pan, Conghui Xu, Juntao Li, Hong Yu, Jennifer Chase, Edward R. Atwill, Xunde Li, Kuncai Chen, Shouyi Chen

**Affiliations:** 1Guangzhou Center for Disease Control and Prevention, Qide Road No.1, Jiahewanggang, Baiyun District, Guangzhou 510440, China; xhcwei@ucdavis.edu (X.W.); gzcdc367@163.com (S.H.); panxh1987@163.com (X.P.); yeast-27@163.com (C.X.); jtwl1011@163.com (J.L.); yu_hong66612@hotmail.com (H.Y.); 2Western Institute for Food Safety and Security, University of California Davis, Davis, CA 95618, USA; jchase@ucdavis.edu (J.C.); ratwill@ucdavis.edu (E.R.A.); xdli@ucdavis.edu (X.L.); 3Department of Population Health and Reproduction, University of California Davis, California, Davis, CA 95616, USA

**Keywords:** strawberry, *E. coli*, total coliform, *Cryptosporidium*, wildlife, water, U-Pick

## Abstract

This study quantified the association of rodent fruit damage and the microbiological quality of irrigation water on the risk of microbiological contamination of strawberries collected from 18 U-pick farms across five different districts in the Guangzhou metropolitan region of southern China. Fifty-four composite strawberries samples, with or without evidence of rodent or avian foraging damage (i.e., bitten), along with 16 irrigation water samples, were collected during the spring of 2014 and winter of 2015 from our cohort of 18 farms. Composite strawberry samples and irrigation water were analyzed for total coliforms, *E. coli*, *Salmonella*, *E. coli* O157, *Giardia*, and *Cryptosporidium*. Total coliforms and *E. coli* were detected in 100% and ~90% of irrigation water samples, respectively. In contrast, *Cryptosporidium* was detected in only two water samples, while *Salmonella*, *E. coli* O157, and *Giardia* were not detected in any water samples. Strawberries with signs of being bitten by wildlife had significantly higher concentrations of total coliforms and *E. coli,* compared to strawberries with no physical evidence of rodent damage (*p* < 0.001). Similarly, *Cryptosporidium* was detected in 7/18 (39%) of bitten, 4/18 (22%) of edge, and 5/18 (28%) of central strawberry samples, respectively. Concentration of *E. coli* on strawberries (*p* < 0.001), air temperature (*p* = 0.025), and presence of *Cryptosporidium* in irrigation water (*p* < 0.001) were all associated with the risk of *Cryptosporidium* contamination on strawberries. *Salmonella* and *Giardia* were detected in <4% strawberry samples and *E. coli* O157 was not detected in any samples. These results indicate the potential food safety and public health risks of consuming unwashed strawberries from U-pick farms, and the need for improved rodent biosecurity of U-pick strawberry fields and enhanced microbiological quality of irrigation water used at these facilities.

## 1. Introduction

Foodborne outbreaks linked to consumption of fresh produce have been reported in numerous countries, such as Norway, Sweden, United Kingdom, USA, and Australia [1]. A recent (2018) outbreak of *E. coli* O157:H7 infections occurred in the USA, linked to the consumption of romaine lettuce from the Yuma growing region—210 people (five deaths) were infected in 36 states [2]. The causative agents of foodborne microbial illnesses associated with the consumption of fresh fruits and vegetables include bacteria, protozoa, and viruses [3]. Among the many commodities of produce consumed raw, strawberry is a popular fruit with increasing consumption and growing demand [4]. However, given that strawberries are typically not washed prior to distribution for retail, this fruit can be a vehicle of foodborne transmission of commensal and pathogenic bacteria, protozoa, and other enteric pathogens [5]. Strawberries are typically cultivated in open fields close to the ground, which makes these fruits vulnerable to a diversity of biological sources and modes of microbial contaminations, compared to those grown in a closed environment (such as greenhouses) or grown above the ground [6,7]. As a consequence, the risk of microbial contamination and foodborne illness is of additional concern if strawberries are consumed raw or with minimal processing [8] when harvested from open fields such as the popular U-pick farms, which are open to the public, and allow consumers to enter the field and directly harvest fruit.

Known and potential sources of on-farm contamination of fresh produce include irrigation water, wildlife intrusion to production fields, human handlers, domestic animals, and biological soil amendments [6,9,10]. For example, a study in Belgium reported that *E. coli* was detected in 71% (40/56) of ponds used for produce irrigation [11]. Among central coastal California farms, both *Salmonella* and *E. coli* O157:H7 were found in water supplies used to irrigate various leafy green commodities [12]. Using such irrigation water contaminated with bacterial pathogens could significantly elevate the risk of microbial contaminations of fresh fruits and vegetables, especially if irrigation water comes in contact with the edible portion of the plant. Similar to concerns regarding irrigation water, wildlife intrusion into production fields can function as a source of contamination of pre-harvest produce. For example, *E. coli* O157 isolated from feral swine matched the strain associated with the 2006 outbreak associated with consumption of baby spinach [13]. More recently, deer mice (*Peromyscus maniculatus*) trapped in produce production fields in central coastal California were found to shed high concentrations of multiple genotypes of *Cryptosporidium* and *Giardia*, along with low levels of *Salmonella* and *E. coli* O157:H7 [14,15].

With the goal of promoting the microbiological safety of fresh fruits and to assist farmers develop good agricultural practices, the current study assessed the microbiological safety of strawberries from U-pick farms in the Guangzhou region and analyzed the association between wildlife intrusion and irrigation water quality on the microbiological safety of this popular fruit.

## 2. Materials and Methods

### 2.1. Strawberry Field Selection and Fruit Collection

Eighteen strawberry fields were enrolled from five districts located from north to south of the Guangzhou metropolitan area, China, for a cross-sectional study during March to April 2014 and January 2015. These strawberry fields allowed consumers to enter the field and directly harvest fruit, typically designated as U-pick farms that are open to the public. Each farm was visited once during the study period to collect samples of fruit between 10 am and 1 pm. Three categories of strawberry were collected from each field: intact strawberries from the center of the field that were often preferred by consumers (herein labeled as central strawberries); intact strawberries from the outer edges of the field that were also preferred by many consumers (herein labeled as edge strawberries); and strawberries exhibiting physical damage consistent with rodent bites or avian pecking from anywhere in the field, but with a tendency to be located at the field edges (herein labeled as bitten strawberries). The distribution of the three categories of strawberries in the field are shown in Figure 1. Strawberries were collected by hand with sterile gloves and placed into sterile zip bags. In addition to strawberry samples, we also collected water samples from ponds inside the farms if the farmers claimed to use the water to irrigate the strawberry fields. Ten L of water was collected into sterile 10-L carboys, placed on ice in coolers, transported to the laboratory within two hours of collection, and stored in a cold room (4 °C) until processed.

The following additional information was collected: farm address, square meter of strawberry fields, ambient air temperature at time of sampling, irrigation method, and frequency. The turbidity of irrigation water was measured using a turbidity meter (Hach Chemical Co., Loveland, CO, USA).

### 2.2. Microbiological Analysis of Strawberries and Irrigation Water

Approximately 60 g (five to six pieces) of strawberries was weighed and placed into a plastic filter bag containing 255 mL of 1 M glycine (pH 5.5) [16,17]. After shaking by hand for 30 s at room temperature, the strawberries were removed and the eluent poured into a centrifuge bottle [16,17]. For irrigation water, 55 mL of each sample was tested for bacteria and the remainder of the 10 L sample was concentrated to approximately 200 mL of retentate using hollow fiber ultrafiltration [18] to test for *Giardia* and *Cryptosporidium*.

To detect *Giardia* and *Cryptosporidium* in strawberry and irrigation water samples, 200 mL strawberry eluent and 200 mL retentate of irrigation water were centrifuged at 4000 rcf at 25 °C for 15 min. The supernatant was discarded, and immunomagnetic separation of (oo)cysts was conducted on 3 mL of pellet, according to the manufacturer’s instructions (Invitrogen Dynal AS, Oslo, Norway) [17]. Immunofluorescent staining of the final IMS products (approximately 50 μL) was performed using the Aqua-Glo G/C Direct kit (Waterborne, New Orleans, LA, USA), according to the manufacturer’s instructions. The slides were examined for (oo)cysts using immunofluorescent microscopy at 200 to 400× amplifications. Concentrations of *Cryptosporidium* and *Giardia* were calculated and expressed as number of (oo)cysts per gram of strawberry and per liter of irrigation water.

We used China’s National Standard (GB) protocol of National Food Safety Standard (Ministry of Health the People’s Republic of China) to detect total coliforms, *E. coli*, *E. coli* O157, and *Salmonella* in the strawberry and irrigation water samples. Specifically, protocol GB4789.3-2010 was used to enumerate total coliforms; GB4789.38-2012 was used to enumerate *E. coli*; GB4789.36-2008 was used to detect *E. coli* O157; and GB4789.4-2010 was used to detect *Salmonella*, respectively. Detailed procedures for detection of these bacteria are described in a previous study [19]. The procedures for detection of these bacteria are briefly described below.

To enumerate total coliforms from strawberry samples, triplicate aliquots of 1.0, 0.1, and 0.01 mL of eluent were re-suspended in tubes filled with 10 mL LST (Lauryl Sulfate Tryptose) broth, incubated at 37 °C for 48 h, and the most probable number (MPN) calculated according to the number of tubes generating positive reactions (bubbles). One loop each of LST from tubes generating positive reactions was transferred into 10 mL *E. coli* broth and incubated at 44.5 °C for 48 h to calculate the MPN of *E. coli*. To detect *E. coli* O157, 25 mL of eluents were added to 225 mL of modified *E. coli* broth and incubated at 37 °C for 24 h. An aliquot of the culture was streaked onto *E. coli* O157 chromogenic medium and incubated at 37 °C for 24 h. Red or purple colonies were transferred to TSI (Triple Sugar Iron) agar and incubated at 37 °C for 24 h. To detect *Salmonella*, 25 mL of eluents was added to 225 mL of Buffered Peptone Water (BPW) and incubated at 37 °C for 18 h. An aliquot of the BPW culture was added to 10 mL of TTB (Tatrathionate Broth) and incubated at 42 °C for 24 h. The culture in TTB was then streaked onto *Salmonella* chromogenic medium and incubated at 37 °C for 24 h. Purple or red colonies were selected for culture on TSI (Triple Sugar Iron) agar at 37 °C for 24 h. Isolates of *Salmonella* and *E. coli* O157 were confirmed biochemically and phenotypically using the VITEK2 GN method (bioMériux, Marcy d’Etoile, France). The same MPN triplicate volumes of original irrigation water (without filtration) and similar procedures were used to enumerate or detect total coliform, *E. coli*, *E. coli* O157, and *Salmonella* from irrigation water samples. Concentrations of bacteria were calculated and expressed as MPN per gram of strawberry and per 100 mL of irrigation water.

### 2.3. Statistical Analysis

For strawberry samples, concentrations of total coliforms and *E. coli* that were greater than 468 MPN/g were treated as equal to 468 MPN/g for statistical analysis. For irrigation water samples, concentrations of total coliforms and *E. coli* that were greater than 11,000 MPN/100 mL were treated as equal to 11,000 MPN/100 mL for statistical analysis. The Chi-squared test was applied to assess whether there is a statistical difference for the positive rates in the three categories of strawberries. The Kruskal-Wallis test was used to determine whether the categories (center, edge, and bitten) of strawberry samples were associated with elevated concentrations of total coliforms and *E. coli* in strawberries. Logistic regression was used to assess the association of *E. coli* concentration on strawberries, air temperature at time of sampling, and presence of *Cryptosporidium* in irrigation water on the risk of detecting *Cryptosporidium* on the surface of the strawberries using STATA 14 software (StataCorp LP, College Station, TX, USA).

## 3. Results

### 3.1. Characteristics of Strawberry Fields

We collected 54 strawberry samples (3 samples per field) and 16 irrigation water samples (one field without a pond, one field sharing a pond with another field) from the 18 enrolled U-pick strawberry fields. The size of the strawberry fields ranged from 2000 m^2^ to 6700 m^2^, irrigated every month to two months using surface drip technology. During sampling days, air temperature at the sampling time in the strawberry fields ranged from 28 °C to 34 °C with a mean of 31 °C during March through April 2014 and 15 °C to 26 °C with a mean of 21 °C during January 2015.

### 3.2. Microbiological Analysis of Strawberry and Irrigation Water

The prevalence of positive samples and concentrations of total coliforms and *E. coli* for the different categories (central, edge, bitten) of strawberry samples are shown in Table 1. Both the prevalence and mean concentrations of total coliforms and *E. coli* were highest in bitten strawberries, followed next by strawberries located at the field’s edge, and lowest in strawberries located centrally. Specifically, the prevalence of detecting total coliforms was 1.3- and 1.5-times higher for bitten strawberries compared to unbitten strawberries at the edge or center of the field, respectively (*p* = 0.05). Similarly, the prevalence of detecting *E. coli* on strawberries was 2.0- and 4.6-times higher for bitten strawberries compared to unbitten strawberries at the edge or center of the field, respectively (*p* = 0.001). The concentration of total coliforms was 4.5- and 7.6 times higher for bitten strawberries compared to unbitten strawberries at the edge or center of the field, respectively (*p* < 0.001). Similarly, the concentration of *E. coli* was 48- and 192-times higher for bitten strawberries compared to unbitten strawberries at the edge or center of the field, respectively (*p* < 0.001). *Salmonella* was detected in only two strawberry samples, both of which were bitten strawberries, and *E. coli* O157 was not detected in any of the samples.

The turbidity of irrigation water ranged from 0.58 NTU to 62.9 NTU with a mean of 34.23 NTU. All the irrigation water samples were found to be positive for total coliforms and 87.5% (14/16) of the water samples were positive for *E. coli*. Concentrations of total coliforms and *E. coli* in irrigation water samples ranged from 30 to >11,000 MPN/100 mL and 0 to >11000 MPN/100 mL, respectively. Approximately 69% (11/16) of the samples had total coliforms concentrations >11,000 MPN/100 mL and 25% (4/16) samples had *E. coli* concentrations >11,000 MPN/100 mL. *Salmonella* and *E. coli* O157 were not detected in any irrigation water samples.

*Cryptosporidium* was detected in 7/18 (39%) of bitten, 4/18 (22%) of edge, and 5/18 (28%) of central strawberry samples (Table 1). The concentrations of *Cryptosporidium* oocysts on strawberry samples ranged from 0 to 0.60 oocysts/g overall, with a mean of 0.01 oocysts/g of bitten, 0.05 oocysts/g of edge, and 0.02 oocysts/g of central strawberry samples. Only two strawberry samples from different fields were positive for *Giardia*: one from edge strawberries and the other one from bitten strawberries, with both of these samples also positive for *Cryptosporidium*. The concentrations of cysts in the two samples were both 0.04 cysts/g. Two irrigations water samples tested positive for *Cryptosporidium* oocysts, with concentrations of 0.1 and 0.2 oocysts/L. Four out of six strawberry samples from these two fields with positive *Cryptosporidium* irrigation water were found to be positive for oocysts. *Giardia* and *Salmonella* was not detected in any irrigation water samples.

Factors significantly associated with the presence of *Cryptosporidium* on strawberries were the *E. coli* concentration on strawberries, air temperature, and presence or absence of *Cryptosporidium* in irrigation water (Table 2). Interpretation of the model is as follows: as the concentration of *E. coli* increases by 10 MPN/g on the surface of the strawberry, the odds of detecting *Cryptosporidium* increase multiplicatively by 1.083 or 8.3% ($e^{10\times 0.008}$ = 1.083). As the air temperature increases per degree Celsius, the odds of detecting *Cryptosporidium* increase multiplicatively by 1.46 or 46% ($e^{1\times 0.378}$ = 1.46). Lastly, the risk of detecting *Cryptosporidium* on strawberries increases almost 8-fold ($e^{1\times 2.053}$ = 7.79) for U-pick farms that had *Cryptosporidium* detected in their irrigation supplies. Figure 2 shows a graphical representation of the model’s predictions of the risk of detecting *Cryptosporidium* on strawberries from U-pick farms as a function of *E. coli* contamination on strawberries at three different air temperatures (19, 27, 33 °C), which are the quartiles of air temperature data for this region of Guangzhou during the sampling periods. In addition, the risk profiles shown in Figure 2 are for U-pick farms that had *Cryptosporidium* detected in their irrigation water. Figure 3 highlights the modeled effect of detecting *Cryptosporidium* in farmer’s irrigation water supplies on the risk of *Cryptosporidium* contamination of strawberries from U-pick farms across a range of concentrations of *E. coli* on strawberries, with air temperature set at 28.5 °C.

Together, these analyses and figures indicate that strawberries picked during hotter seasons from farms with higher levels of *E. coli* on their strawberries and *Cryptosporidium* in their irrigation water are at considerable risk (probability >75%) of being contaminated with this protozoal parasite.

## 4. Discussion

Wildlife intrusion in production fields is a risk factor for pre-harvest produce contamination [10]. During sampling, we consistently observed evidence of wild animals (rodents, snakes, and lizards) in different strawberry fields, demonstrated by their tracks or scat and physical damage from foraging on the fruit. Compared to central strawberries, edge strawberries may be at a higher risk of coming into contact with wildlife feces or being partially bitten due to terrestrial wildlife often using the furrow for ingression into the field. Both the prevalence and concentration of the two indicator organisms—total coliform and *E. coli*—on bitten strawberries were significantly higher than in other locations or non-bitten strawberries in our study (Table 1). Additionally, the prevalence of *Cryptosporidium* oocysts was highest in bitten strawberries and the only two *Salmonella* positive samples were also detected in bitten strawberries. Our results suggest that strawberries with evidence of animal contact or being bitten had higher microbiological risk; hence, these results suggest that visitors to U-pick farms in the greater Guangzhou region should not select strawberries with such signs of animal contact and farmers should not harvest these strawberries for human consumption.

For central and edge strawberries, the mean (0.4 and 1.6 MPN/g) and maximum (4 MPN/g and 10 MPN/g) concentrations of *E. coli*, respectively, were near the standard concentrations of *E. coli* specified in microbiological quality guides for ready-to-eat food (fruits and vegetables) in England [20], Australia and New Zealand [21], and Korea [22], which were <20 CFU/g, 3 CFU/g, and 0 CFU/g, respectively. Currently there is a lack of a national microbiological standard for ready-to-eat fruits and vegetables in China [22], but the findings of this study, combined with existing standards from other countries, could provide useful information for Chinese public health authorities when developing regulations for the microbiological safety of fresh fruits.

The risk of *Cryptosporidium* contamination on strawberries was significantly associated with *E. coli* levels on the strawberries (Table 2, Figure 2 and Figure 3). Our results were consistent with the assertion that *E. coli* can function as a crude indicator of fecal contamination of fresh produce, despite several genera of coliforms being common non-fecal contaminants [23,24]. Additional studies have used *E. coli* as an indicator of potential pathogen contamination of irrigation water, including *Cryptosporidium parvum* [25]. Given that the risk of *Cryptosporidium* contamination of strawberries was substantially higher for fruit from U-pick farms with detectable *Cryptosporidium* in their irrigation water supply and for fruit with higher levels of *E. coli* concentration, similar studies in the future should attempt to identify: (1) the source of fecal contamination (humans vs. animals) and (2) the species/genotypes of *Cryptosporidium* in wildlife (e.g., rodents) in surrounding habitats and as the source of microbiological contamination of irrigation water. Previous studies have mentioned that the occurrence of *Cryptosporidium* in irrigation water may be a risk of protozoal contamination of fresh produce [26,27]. Unfortunately, the low concentration of oocysts found on the strawberries and in the irrigation water prevented us from molecular genotyping and determining the species of *Cryptosporidium* in these strawberry fields.

The risk of *Cryptosporidium* contamination of U-pick strawberries was considerably higher during the hotter seasons, compared to the cooler ones (Figure 2). During this study, we took samples across two seasons—spring (average air temperature was 31 °C) and winter (average air temperature was 21 °C)—in the greater Guangzhou region. Given the large number of collinear factors that may be associated with *Cryptosporidium* contamination, we can only speculate on the underlying causal mechanism driving this relationship. For example, spring is the rainy season in Guangzhou, with 18 and 14 days of precipitation in March and April 2014, respectively. In contrast, there were only 5 days with precipitation in January 2015 (winter season). Numerous studies have found an association between rainfall and the prevalence or concentration of *Cryptosporidium*, such as in river water [28], in surface water used for irrigation [29], in wildlife [30], and in produce [31]. Our observations are consistent with findings in these reports and indicate a high microbiological risk in U-pick farms in rainy and hotter seasons. Because precipitation is a strong seasonal driver for cryptosporidiosis in moist tropical locations [32], improvement in regulatory policies and management practices is needed in order to minimize the food safety and public health risks associated with the consumption of strawberry from U-pick farms.

## 5. Conclusions

To the best of our knowledge, this work is the first survey studying the impact of wildlife intrusion and irrigation water quality on the microbiological safety of fresh strawberries at U-pick farms in the Guangzhou region. Findings from this study suggest that management practices, such as improved irrigation water quality and enhanced pest control measures may reduce the microbiological contamination of fresh strawberries and prevent wildlife contact with the fruit. Future studies should focus on the source(s) of microbiological contamination of irrigation water and the exact species of rodent or wildlife feeding or coming into contact with the fruit, along with the species and genotype of pathogens in surrounding habitats, in an effort to maintain or improve food safety for such a popular fruit.

## Figures and Tables

**Figure 1 ijerph-16-04910-f001:**
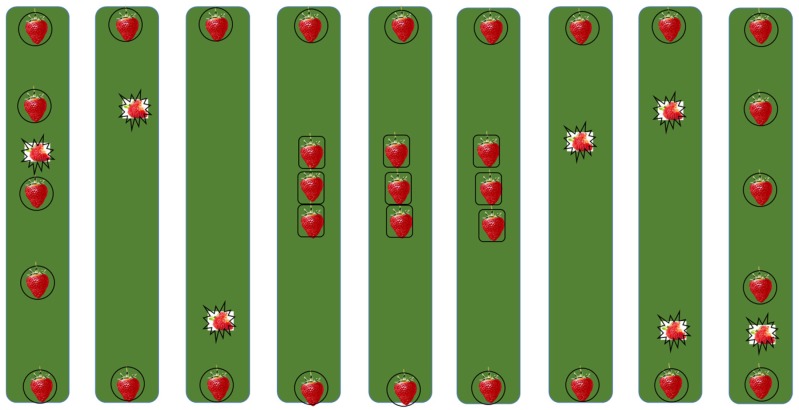
The types and distribution of samples collected in strawberry fields. Note: The rectangle marked strawberries (central strawberries) represent intact strawberries from the center of the field that were often preferred by consumers. The circle marked strawberries (edge strawberries) represent intact strawberries from the outer edges of the field that were also preferred by many consumers. The star marked strawberries (bitten strawberries) represent strawberries that were bitten by animals from surrounding habitats. The green stripes represent planting beds and the white background represents the furrow.

**Figure 2 ijerph-16-04910-f002:**
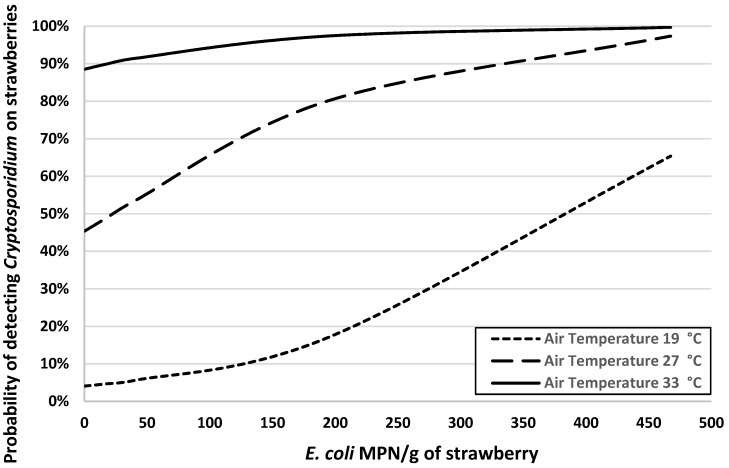
Logistic regression model for the probability of *Cryptosporidium* contamination of strawberries at U-pick farms in Guangzhou, China, as a function of air temperature and the concentration of *E. coli* on strawberries in farms with *Cryptosporidium* detected in their irrigation water supply.

**Figure 3 ijerph-16-04910-f003:**
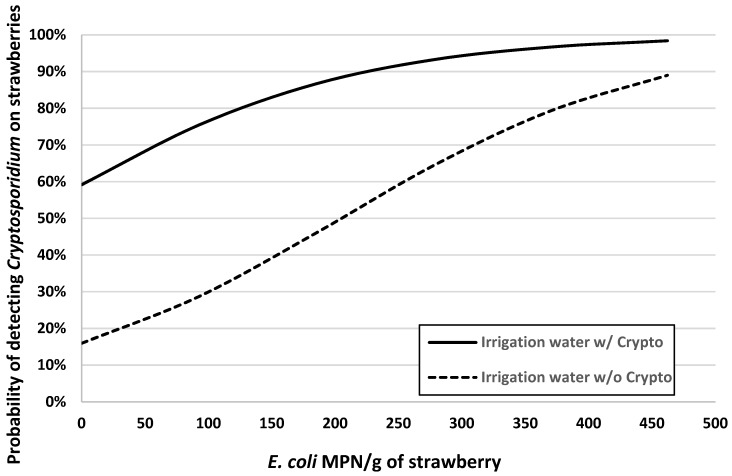
Logistic regression model for the probability of *Cryptosporidium* contamination of strawberries at U-pick farms in Guangzhou, China, as a function of concentration of *E. coli* of strawberries and the presence or absence of *Cryptosporidium* in irrigation water supplies. Air temperature was modeled as constant at 28.5 °C.

**Table 1 ijerph-16-04910-t001:** Prevalence and concentrations of total coliforms and *E. coli* for strawberries located centrally, at the edge of the field, or with evidence of wildlife damage (bitten) at U-pick farms within the Guangzhou metropolitan region, China.

Sample Category		Bacterial Prevalence and Concentration		
	Parameters	Total Coliforms	*E. coli*	Prevalence of *Cryptosporidium*
Central strawberries	% (n/n) positive	61% (11/18)	17% (3/18)	28% (5/18)
	Mean (MPN/g)	45.0	0.4	0.02
	Range (MPN/g)	0–196	0–4	0–0.13
Edge strawberries	% (n/n) positive	72% (13/18)	39% (7/18)	22% (4/18)
	Mean (MPN/g)	74.5	1.6	0.05
	Range (MPN/g)	0–468	0–10	0–0.60
Bitten strawberries	% (n/n) positive	94% (17/18)	78% (14/18)	39% (7/18)
	Mean (MPN/g)	341.3	76.7	0.01
	Range (MPN/g)	0–468	0–468	0–0.06

Note: The prevalence of total coliforms and *E. coli* were significantly associated with category of sample (*central*, *edge*, *bitten*), *p* = 0.05 and *p* = 0.001, respectively. Concentrations of total coliform and *E. coli* were significantly higher for bitten strawberries compared to unbitten located centrally or at the edge of the field (*p* < 0.001).

**Table 2 ijerph-16-04910-t002:** Logistic regression model for the presence of *Cryptosporidium* on strawberries with independent variables *E. coli* concentrations of strawberries, air temperature, and presence of *Cryptosporidium* in irrigation water in U-pick farms in Guangzhou, China.

Independent Variables	Coefficient	*p*-Value	95% CI (Confidence Interval)	Odds Ratio
*E. coli* concentrations of strawberries (MPN/g)	0.008	<0.001	0.004, 0.013	1.008
Air temperature (°C)	0.378	0.025	0.048, 0.708	1.459
*Cryptosporidium* in irrigation water	2.053	<0.001	0.926, 3.180	7.790
Constant	−12.451	0.012	−22.134, −2.767	3.92 × 10^−6^

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
