# Peer review of "Microbiological Contamination of Strawberries from U-Pick Farms in Guangzhou, China"

_ijerph, 2019, doi:10.3390/ijerph16244910_

Round 1

Reviewer 1 Report

Thank you for the interesting and well-written manuscript.

Maior points:
You should explain what "U-Pick" means much earlier in the manuscript. I was confused from the title until the explanation in "Materials and Methods" of the meaning of this word. It should at least be explained early in the introduction section.
Line 142-145: Why did you truncate the concentrations at 468 MPN/g resp. 11,000 MPN/100 mL?
Line 166 and later: It would be good with confidence intervals for the ratios (e.g. a CI for 1.3- times higher)
Line 206: "increase 780%" should be "increase 680%" to be consistent with the other results.
Line 247-251: It is unclear how concentrations as CFU/g from the liytterature are related to your concentrations in MPN/g.

Minor points:
Line 45: "close the ground" should maybe be "close to the ground".
Line 48: full stop is missing after [6,7]
Line 89: Something of the formating is wrong with this line
Figure 1: The resulution of this figure is not very high.
Figure 1, figure text: "represent for" should just be "represent"
Table 1: Virtual separation (either a bit extra space or a line) between the categories would make the table more readable.
Line 244: "those" should be removed
Line 301: "Its'"should be "its"

Author Response

Thanks for the valuable comments and suggestions!

Maior points:

Point1: You should explain what "U-Pick" means much earlier in the manuscript. I was confused from the title until the explanation in "Materials and Methods" of the meaning of this word. It should at least be explained early in the introduction section.

Response 1: added U-pick explanation in the introduction line50-51.

Point2: Line 142-145: Why did you truncate the concentrations at 468 MPN/g resp. 11,000 MPN/100 mL?

Response 2: Great question! We can’t analyze the data with the values, which are greater than a number. We only have a few of these kinds of values. Eventually, we choose this solution without influencing the results. It would be best to get the exact number.

Point3: Line 166 and later: It would be good with confidence intervals for the ratios (e.g. a CI for 1.3- times higher)

Response 3: Yes, CI will be a better way to describe it and provide more details. However, we can’t calculate CI here because we only compare the prevalence of total coliforms between bitten strawberries and central/edge strawberries. The same for E. coli.

Point4: Line 206: "increase 780%" should be "increase 680%" to be consistent with the other results.

Response 4: Corrected.

Point5: Line 247-251: It is unclear how concentrations as CFU/g from the liytterature are related to your concentrations in MPN/g.

Response 5: These are two major methods to quantify the concentration of bacteria. Many publications have been published to justify that the results of the two methods are comparable.

Minor points:

Point6: Line 45: "close the ground" should maybe be "close to the ground".

Response 6: Corrected.

Point7: Line 48: full stop is missing after [6,7]

Response 7: Corrected.

Point8: Line 89: Something of the formating is wrong with this line

Response 8: Corrected.

Point9: Figure 1: The resulution of this figure is not very high.
            Figure 1, figure text: "represent for" should just be "represent"

Response 9: Corrected.

Point10: Table 1: Virtual separation (either a bit extra space or a line) between the categories would make the table more readable.

Response 10: Changed.

Point11: Line 244: "those" should be removed 

Response 11: Corrected.

Point12: Line 301: "Its'"should be "its"

Response 12: Corrected.

Reviewer 2 Report

It’s of great interest to investigate the health safety aspects of fresh fruit, this manuscript provide a first survey on the impacts of wildlife intrusion and irrigation water quality on the microbiological safety of fresh strawberries at U-Pick farms in Guangzhou region. For the research that improved irrigation water quality and improved pest control measures such as rodent traps may reduce microbiological contaminations of fresh strawberries and prevent wildlife intrusion and contact with the fruit.

However, the sources of microbiological contamination in irrigation water and the exact species of rodent or wildlife contacting the strawberries, along with species and genotypes of pathogens in wildlife from the surrounding habitats needs to be explained for food safety prevention. What are the approaches the authors would like to do?

In general, the manuscript is well designed, the materials and methods are well presented, the presentation of the results together with the discussion bring clear information to the readers. Therefore, I recommend the publication of this manuscript.

Author Response

Comments and Suggestions for Authors

It’s of great interest to investigate the health safety aspects of fresh fruit, this manuscript provide a first survey on the impacts of wildlife intrusion and irrigation water quality on the microbiological safety of fresh strawberries at U-Pick farms in Guangzhou region. For the research that improved irrigation water quality and improved pest control measures such as rodent traps may reduce microbiological contaminations of fresh strawberries and prevent wildlife intrusion and contact with the fruit.

However, the sources of microbiological contamination in irrigation water and the exact species of rodent or wildlife contacting the strawberries, along with species and genotypes of pathogens in wildlife from the surrounding habitats needs to be explained for food safety prevention. What are the approaches the authors would like to do?

In general, the manuscript is well designed, the materials and methods are well presented, the presentation of the results together with the discussion bring clear information to the readers. Therefore, I recommend the publication of this manuscript.

Response 1: Thanks for the valuable comments and suggestions.

In order to get more details and be able to answer these questions, metagenomics can be used to track the sources of microbiological contamination in irrigation water;

Day and night camera can be set up to record the species of the rodents or wildlife around;

Species and genotypes of pathogens in wildlife from the surrounding habitats can be identified by collecting the faces of wildlife around the fields or setting up traps to catch the rodents.
